# Ownership structure and financial performance: Evidence from Kenyan commercial banks

**Peter Njagi Kirimi**[1]*, **Samuel Nduati Kariuki**[1], **Kennedy Nyabuto Ocharo**[2]

**1** Department of Business Studies, School of Business and Economics, University of Embu, Embu, Kenya,
**2** Department of Economics, School of Business and Economics, University of Embu, Embu, Kenya

☉ These authors contributed equally to this work.
* Pnjagi06@gmail.com

**Data Availability Statement:** All relevant data are within the manuscript and its Supporting information files.

**Funding:** The author(s) received no specific funding for this work.

## Abstract

The study examined the relationship between ownership structure and financial performance of commercial banks in Kenya for the period 2009–2020. The data were collected from audited financial statements of 39 commercial banks in Kenya. Regression results found strong evidence on ownership structures in explaining the differences in commercial banks' financial performance. The results established that the greatest influence of ownership structures was on net interest margin at 53.04% and return on assets at 31.37%. Influence of ownership structures was found to be low on return on equity at 3.32% and earnings per share at 2.13%. The results found a negative association between state ownership and net interest margin, negative association between management ownership and both net interest margin and earnings per share, negative association between institutional ownership and return on assets and a negative association between foreign ownership and earnings per share. Based on the findings, commercial banks should vary their ownership structures to boost financial performance. Secondly, banks with high percentage of state ownership should consider partial privatization to improve corporate governance practices. Third, banks should adopt managerial ownership policy limiting the proportion of equity stock on executives to limit their powers in strategic decision making. Fourth, the study proposes a percentage limit on equity stock of an individual institutional investor. Lastly, the study proposes that bank's management to come up with a policy detailing the role and place of foreign investors in strategic decision making to ensure their presence in every decision undertaken by bank managers.

## 1. Introduction

Firms use many tools to enhance and promote their performance. Ownership structure is one of the major tools applied to enhance performance in many firms across the globe [1]. To enhance financial planning in many firms, ownership structure is a vital component and is used by many firms' directors [2] since any firm's board efficiency hangs on the overall diversity of the ownership structure [3]. Ownership structure represents a powerful means for the

**Competing interests:** The authors have declared that no competing interests exist.

firm's management to exercise control to improve performance [4]. This is because different ownership structures can be applied in countering the agency problems, thereby boosting the relationship between the firm's management and other stakeholders of the firm.

Globally, the relationship between ownership structure and financial performance has been of primary concern to scholars, management, academics, policy makers and investors for decades. This concern comes from the fact that ownership structure influences firm's corporate governance on major decisions thereby impacting on the firms' financial performance. The authors in [5] argued that this relationship depends on the various types of ownership structures that exist in different firms. Ownership structure and performance is a contentious issue that has been a subject of debate in all businesses globally [6]. Ownership structure reveals the distribution of equity in a firm and the identity of the equity owners [7]. Board decisions are influenced by corporate governance mechanisms which are based on ownership structures in place by firms. Ownership structures lead to agency problems resulting from conflicts between management and shareholders. As noted by [8], this conflict lowers the value of the firm when managers put their interest before those of the owners.

Ownership structure of a firm plays a great role in decision making and cost control [9]. The form of ownership structure in a bank dictates and provides the level of influence and control one has on management of the bank in making vital decisions. Ownership structure in banks influences corporate governance, corporate strategy and performance [10]. In the context of the agency theory, the study explored a critical question, whether ownership structure is a key component in enhancing financial performance. This question impelled the study to focus on two expectations. The first was to establish the role of ownership structure in enhancing the firm's performance. The second was to fill the related gap in literature relating to the relationship between ownership structure and financial performance, especially in the Kenyan banking context.

This paper contributes to literature in many ways. First, the study investigates the relationship between ownership structure and financial performance of commercial banks in Kenya, a developing market economy considered as overbanked. This was due to the fact that there is no agreement from empirical results on the relationship between ownership structure and banks' financial performance. For example, authors in [11] found that ownership structure had no influence on performance. In another study [12] authors found evidence that ownership structure positively determined Asian banks' profitability. Moreover, it was established that ownership structure impacted on Chinese banks' overall performance positively [13]. This study aims at solving this controversy by presenting evidence from a developing economy through investigating the relationship between ownership structure and financial performance using data from the commercial banking sector in Kenya.

Secondly, Kenya ranked very low in provision of investor protection compared to other developing and developed market economies globally. The strength of the last investor protection index carried out in Kenya stood at 5.3% [14]. The findings would benefit policymakers by assisting bank management to enhance financial performance through modification of ownership structure to influence governance decisions and investor protection policies. Thirdly, to the best of our knowledge, there is no study that has investigated the effect of ownership structure on banks' financial performance in Kenya using a wider scope of four financial performance measures. The finding of this study therefore equips policymakers and lawmakers with ownership structure knowledge required to formulate policies for improving financial performance in commercial banks.

The remainder of this paper is organized as follows: section 2 presents the background, section 3 discusses theoretical and empirical literature review and hypotheses development.

Section 4 presents the methodology and section 5 presents empirical results and discussion. Finally, section 6 presents the conclusions, recommendations and suggestions for future study.

## 1.1 Background

Kenya like other emerging economies is classified as developing. Through the decades Kenya has walked through the path to a free market economy with economic policies aimed at improving investment opportunities. Due to this, the Kenyan economy has witnessed an increase in number of banks which has led Kenya being classified as one of the overbanked developing economy. To ensure that the Kenyan banking sector remained financially sound, the Kenyan government came up with mechanisms to monitor banking operations through the Central Bank of Kenya (CBK) which emphasized on adherence to BASEL Accords that lays down measures to ensure minimum liquidity and capital requirements are met. The CBK also came up with prudential guidelines to protect depositor's funds. For example, the passing of the 1989 Banking Act introduced banking regulations that govern commercial banks' operations after 9 banks had collapsed. However, this did not solve the problem as six more banks collapsed before 1998. Between 2007–2015, four banks, that is, Kenya Finance Corporation, Trade bank, Euro bank and Charter House bank collapsed. The collapse of these banks was mainly attributed to poor governance structures which led to a decline in financial performance [15, 16]. In addition, between August and October 2015, Dubai bank and Imperial bank were put under statutory management.

In the last decade, Kenya has witnessed an increased rate of mergers and acquisitions in the banking sector with recent cases witnessed in years 2019 and 2020. For example, the merger and acquisition between Commercial Bank of Africa Limited (CBA) and NIC Group PLC was completed in September, 2019. Kenya Commercial Bank acquired 100 percent shareholding of National Bank of Kenya in September, 2019. In the year 2020, Access Bank PLC acquired Transnational Bank PLC and Commercial International Bank of Egypt acquired 51 percent shareholding of Mayfair Bank Ltd. During the same year Kenya Commercial Bank acquired part of Imperial Bank's assets which were valued at KES 3.2 billion and assumed liabilities of the same value. Still in the year 2020, the Co-operative Bank of Kenya Ltd acquired 90 percent equity of Jamii Bora Bank.

The mergers and acquisitions were influenced by the decline in banks' performance resulting from ineffective governance mechanisms that were put into place by the management. For example, the CBK's annual supervision report noted that the banking industry experienced a sharp decline on its profitability by 9.6% in the year 2017 [17]. Later in the year 2019, the pre-tax profits of the sector decreased by 29.3% from KES 159.1 billion in December 2019 to KES 112.1 billion in December 2020. The decrease in profitability was attributed to a higher growth in total expenses.

With erosion of consumer confidence in small and mid-sized banks, large commercial banks that control an estimated 80% of the market in Kenya could become even more dominant by acquiring the businesses of small and mid-sized banks that face governance challenges leading to poor financial performance [18]. Majority of banks in developing economies including Kenya have issues with corporate governance due to ownership structures. These governance issues range from inadequate protection of banks' investors, management transparency problems to agency problems. These issues are due to lack of adequate oversight and consistency in rules, leading to many corporations suffering market deficiency [19]. Firms are unable to sustain their market operations and access additional capital. In addition, investments by minority equity holders would be affected by large shareholders who influence and have authority to influence decisions. Therefore, this study anticipated that ownership structure as

corporate governance tool, would play a vital role in influencing banks' financial performance through shielding equity holders' investments.

## 2. Theoretical literature review

### 2.1 The agency theory

The agency theory which is widely used as the basis for corporate governance studies dates back to the work of [20] and later developed by [9]. The theory suggests that there exists an agency problem between the principal and the agent which leads to agency costs. This dilemma exists where managers acting as agents of equity holders are motivated to act in their own best interests, which are contrary to those of the owners [21]. In most cases, managers are influenced to act contrary to owners' expectations which affect firms' financial performance [22]. To counter the agency problem, both shareholders' and managers' interests should be aligned with the objectives of the firm. For better firm's performance, reference [23, 24] suggest provisions of effective ownership structures that are aimed at lowering agency costs through control of management decisions. Agency problems are themselves motivators of ownership structures in many firms [25] and therefore act as the turn point for corporate governance debates among board of directors and firms' management. A study by [26] found that ownership structure in firms helps to reduce agency costs by regulating how a company is managed. Other studies proposed the refraining approach such as assigning a portion of shareholding to top managers and setting up of independent audit committees [27–30]. This approach is expected to help align manager's interest with that of owners as well as engaging in beneficial and profitable business activities.

### 2.2 Empirical literature and hypothesis development

Commercial banks have a central position and play a great role in the financial system [31]. The operations of banks are put in check by many stakeholders ranging from government regulators, investors as well as bank owners. Banks' operations are as a result of governance structure put into place. An ownership structure as a major tool for determining corporate governance has been a subject of debate by researchers for decades. Ownership structure sets up how equity is distributed in a firm and through this the identity of equity owners and their voting power is revealed [7]. The performance of every firm is affected by various classes of shareholders due to their differences in strategic decision making powers [32]. Before the financial crisis hit business in 2008, ownership structure had played a vital role in boosting up financial performance [33]. In addition [33], provided firm regulators with an understanding of improving corporate governance practices. After the financial crisis of 2008, scholars and researchers across the globe relooked at the relationship between ownership structure and financial performance as ownership structure was viewed as a major tool in enhancing recovery both in the financial and non-financial sectors [23, 26, 34–36]. In today's businesses where separation of ownership and management is emphasized, managers of firms are tasked with the control of all major strategic decisions. In this way they are expected to have implications on the validity of the wealth-maximizing goal of the firm. Commercial banks however, are not exception from this rule. In making strategic financial decisions, bank managers may pursue their interests resulting to poor financial performance. It is for this reason that [6] found that ownership structure and performance is an issue that has been a subject of debate for a long time. Ownership structure plays a vital role in decision making and control [9]. From this early notion, the form of ownership structure in a bank dictates and provides the level of influence and managerial control in critical decisions making especially those pertaining to bank's corporate governance practices. This study sought to investigate the association of ownership

structure on financial performance of commercial banks and focused on four dimensions of ownership structure aimed at reducing the agency problems. These four dimensions formed the basis for hypotheses development.

**2.2.1 State ownership and financial performance.** As posited by [37], state ownership of commercial banks is ineffective due to political interference, weaker managerial incentives compared to privately owned banks and weak monitoring efforts of government. Political interference conflicts with firm objectives of profit maximization resulting to poor financial performance of commercial banks. Prior empirical studies found that state owned banks exhibit lower financial performance associated with weaker corporate governance and risk levels that are very high [38–40]. In addition, Lee and Kim [41] established a negative relationship between state ownership and financial performance in the Korean banking sector. This finding was later confirmed in the Chinese banking sector by [42] who established that poor performance was associated with state ownership. However, these results contradicted the work of [43] who found a positive relation between state ownership and financial performance of commercial banks. A more recent study also found a positive relation between state ownership and financial performance of banks in Yemen [44]. On the other hand, study by [45] found no difference in performance between state-owned banks and privately owned banks in Africa. This finding was later supported by [46]. Empirical evidence from the studies reviewed on the relationship of state ownership on financial performance show that scholars do not agree on a common finding. Therefore, this study sought to examine the relationship between state ownership and banks' financial performance by speculating the following:

$H_{01}$: There is no association between state ownership and financial performance of Kenyan commercial banks.

**2.2.2 Managerial ownership.** Managerial ownership is the proportion of shares that is owned by the executives. Managerial ownership as explained by the agency theory is one of the techniques used to reduce agency problems. According to [9] managerial ownership is applied to improve the value of the firm attained through an increased financial performance. The separation of ownership and control is a subject of concern to many researchers as empirical studies show mixed findings on the relationship between managerial ownership and financial performance. According to the convergence-of-interests hypothesis, an increase in managerial ownership can reduce agency problems drastically [9]. However, the entrenchment hypothesis proposed that a higher level of management ownership in a firm decreases the firm's performance associated with more voting power by executives to control strategic decision-making [47]. Empirical results have established that management ownership has a positive impact on financial performance. For example [32], found that an increase in management ownership positively impacted on firm's performance in Jordan. This finding was also echoed by authors in studies by [36, 48, 49]. A more recent study by [2] agreed with the earlier findings that increasing the executive shareholding increases financial performance. In addition, a study by [50] found that managerial ownership exerted a positive effect on financial performance in Thai firms. Further [51], observed that management ownership is a structure that supports both the interests of shareholders and managers to enhance performance of a firm.

In contrast, other studies supported the entrenchment hypothesis, that is, management ownership negatively relates to firm's performance [47, 52–54]. In addition, studies by [55, 56] found a significant and negative relationship between managerial ownership and commercial banks' financial performance in United States of America and Australia

respectively. Further [57], established a significant negative effect of managerial ownership on bank performance.

Based on the literature reviewed, this study examined the role of management ownership in the performance of the Kenyan commercial banking sector. The following null hypothesis was formulated;

$H_{02}$: Management ownership does not influence financial performance of commercial banks in Kenya.

**2.2.3 Institutional ownership.** Institutional ownership is another tool used to mitigate the impact of agency problems where large individual shareholders make decisions for their own interests at the expense of minority stockholders. Institutional ownership is described as ownership of huge stock in a firm by other institutions [58]. This ownership is associated with high financial performance as a result of high quality management and improved corporate governance [2, 59]. Corporate governance in a firm can be strengthened by institutional ownership [60]. A study by [61] asserts that increasing the portion of institutional ownership prevents managerial fraud. Institutional investors holding a proportion of capital for some time in a firm acquires high level of knowledge that allows them to perform their monitoring role effectively [59]. However, previous empirical studies found mixed conclusions. Some studies established a positive relationship between institutional ownership and firms' financial performance [51, 62–67]. Other studies observed a negative association between institutional ownership and financial performance [68, 69]. On the basis of the literature presented here, the following hypothesis was formulated;

$H_{03}$: Institutional ownership has no relationship with financial performance of Kenyan commercial banks.

**2.2.4 Foreign ownership.** Foreign ownership is another ownership structure known to influence corporate governance in firms. Foreign ownership as applied in the banking sector influences strategic decision making. This is possible through the entry of foreign banks in an economy bringing in efficiency, better capitalization and technical capacity which can also spread to other banking institutions [70]. On the contrary, in case of a financial crisis, foreign ownership through entry of foreign banks in an emerging economy may transfer or trigger a financial crisis that may affect performance of the entire banking sector [71]. Empirical evidence shows that introduction of foreign ownership is beneficial to the firms. For example, authors [13] found that banks with foreign ownership appeared to have better asset quality and overall performance in China. In addition, banks with foreign ownership in Uganda and Botswana performed better compared with their local counterparts [72]. Foreign ownership comes with more experience and knowledge thereby supporting adoption of new corporate governance practices [73]. Foreign ownership act as a salient monitoring tool required to protect firm's profits and shareholders' wealth [74]. In contrast [75], argued that some foreign owners may behave passively in contrary to their monitoring roles for various reasons allowing managers to misrepresent information for their own interests, especially where foreign owners are directly affected by investment duration. In line with these findings, this study hypothesized that;

$H_{01}$: Foreign ownership has no relationship with financial performance of commercial banks in Kenya.

## 3. Methodology

### 3.1 Sample selection

The study focused on Kenyan commercial banks. Data was obtained from annual financial reports and statements from 2009 to 2020. Financial reports and statements provided data which was relevant in relation to the variables in this study. The study used data from 39 commercial banks out of the 42 commercial banks. Three banks were dropped for lack of data for the entire period of study.

### 3.2 Model and measurement of the variables

To meet the purpose of this study, four models were developed to test the relationship between ownership structures and the various financial performance measures employed in the study. The models developed were;

$$NIM_{it} = \beta_0 + \beta_1 SO_{it} + \beta_2 MO_{it} + \beta_3 IO_{it} + \beta_4 FO_{it} + \beta_5 BS_{it} + \beta_6 CR_{it} + e_{it} \tag{1}$$

$$EPS_{it} = \beta_0 + \beta_1 SO_{it} + \beta_2 MO_{it} + \beta_3 IO_{it} + \beta_4 FO_{it} + \beta_5 BS_{it} + \beta_6 CR_{it} + e_{it} \tag{2}$$

$$ROA_{it} = \beta_0 + \beta_1 SO_{it} + \beta_2 MO_{it} + \beta_3 IO_{it} + \beta_4 FO_{it} + \beta_5 BS_{it} + \beta_6 CR_{it} + e_{it} \tag{3}$$

$$ROE_{it} = \beta_0 + \beta_1 SO_{it} + \beta_2 MO_{it} + \beta_3 IO_{it} + \beta_4 FO_{it} + \beta_5 BS_{it} + \beta_6 CR_{it} + e_{it} \tag{4}$$

Where $NIM_{it}$, $EPS_{it}$, $ROA_{it}$, $ROE_{it}$ represent bank 's performance at a given time (t), SO, MO, IO, and FO are abbreviations for state ownership, managerial ownership, institutional ownership and foreign ownership respectively, BS and CR are abbreviations for control variable bank size and credit risk respectively.

Based on the previous studies, the dependent, independent and control variables were measured as follows;

**3.2.1 Dependent variable: Financial performance.** This study used four financial performance measures, that is, net interest margin (NIM) earnings per share (EPS), return on assets (ROA) and return on equity (ROE). ROA and ROE are traditional measures that have been used in many empirical studies [76, 77]. NIM has been used as a measure of financial performance in financial firms [78–80]. In addition, study by [81, 82] confirm EPS as an appropriate measure of firm's financial performance as it summarizes the firm's earning ability.

**3.2.2 Independent variables: Ownership structure and control variables.** The study considered four categories of ownership structure in line with existing literature; state ownership which represents a proportion of firm's stake held by the state, managerial ownership which represents the percentage of equity of shares held by top managers of a firm [36], institutional ownership which represents a proportion of equity shares held by institutional investors [23] and finally, foreign ownership which represents a percentage of shares held by foreign entities or individuals [73].

The study further adopted bank size and credit risk as control variables. The size of the firm indicates the muscle stretch of a firm to fund and organize its operations [83]. In this study, total bank assets were used as a measure of bank size. The second control variable was credit risk, which is a ratio of non-performing loans (NPLs) to total loans and advances. NPLs affect banks' financial performance negatively by denying banks interest income. Table 1 shows a summary of the variables that were used in the model.

**Table 1. Operationalization and measurement of variables.**

| Variable | Abbreviation | Measurement |
|---|---|---|
| Independent variables: | | |
| Ownership Structures: | | |
| (i) State ownership | SO | (i) Percentage of bank's equity shares held by state |
| (ii) Managerial ownership | MO | (ii) Percentage of bank's equity shares held by managers |
| (iii) Institutional ownership | IO | (iii) Percentage of bank's equity shares held by institutions |
| (iv) Foreign ownership | FO | (iv) Percentage of bank's equity shares held by foreign individuals and institutions |
| Control variable: | | |
| (i) Bank size | BS | (i) Natural log of Total assets |
| (ii) Credit risk | CR | (ii) NPLs/Total loan and advances |
| Dependent variables: | | |
| (i) Net Interest Margin | NIM | (i) Net interest income to total assets |
| (ii) Earnings per Share | EPS | (ii) Net income to shares outstanding |
| (iii) Return on Assets | ROA | (iii) Net income to total assets |
| (iv) Return on Equity | ROE | (iv) Net income to total equity |

# 4. Empirical results and discussion

## 4.1 Descriptive statistics

Descriptive statistics for the study variables are shown in Table 2. The net interest margin was 0.064. This demonstrated that banks' performance in terms of NIM was relatively low deviating with 0.219. In addition, NIM varied from a minimum of -0.021 to a maximum of 4.715. This low performance implied that the entire commercial banking sector performance could be rated low during the period under study. The mean EPS was 35.564 with a minim of -186.35 and a maximum of 727.22. These statistics indicate a wide variation in EPS with a standard deviation of 100.07 in the Kenyan context. This an indication that there was a significant difference within commercial banks in Kenya in terms of dividend payments. The mean value of ROA was 0.017 with a minimum of -0.247 and a maximum of 1.416. These statistics indicate a relatively low performance with a standard deviation of 0.073. These results of low performance in ROA led to the conclusion that there was underperformance of banks' assets in generating revenue. The mean of ROE was 0.377. This statistic implies that banks' performance in

**Table 2. Descriptive statistics.**

| Variable | Mean | Std. Dev. | Min | Max |
|---|---|---|---|---|
| SO | 0.57 | 0.198 | 0 | 0.908 |
| MO | 0.141 | 0.250 | 0 | 1 |
| IO | 0.585 | 0.348 | 0 | 1 |
| FO | 0.345 | 0.409 | 0 | 1 |
| BS | 7.567 | 0.697 | 5.143 | 9.481 |
| CR | 0.145 | 0.279 | 0 | 4.523 |
| NIM | 0.064 | 0.219 | -0.021 | 4.715 |
| EPS | 35.564 | 100.07 | -186.35 | 727.22 |
| ROA | 0.017 | 0.073 | -0.247 | 1.416 |
| ROE | 0.377 | 5.400 | -0.948 | 115.423 |

ROE was moderate though the variation was huge reporting a minimum of -0.948 and a maximum of 115.423.

The results in Table 2 show that there was some diversity in terms of banks' ownership structure. The mean state ownership was 0.57 while that of managerial ownership was 0.141. These statistics demonstrate a considerably low managerial ownership in Kenyan banks as compared to state ownership. The mean institutional ownership was 0.585. This demonstrated a considerable institutional ownership in a number of commercial banks in Kenya. As a result, we conclude that this ownership structure was beneficial in many commercial banks in Kenya through offering checks on managerial decisions. The mean of foreign ownership was 0.345 with a standard deviation of 0.409. This showed existence of a considerable foreign ownership in Kenyan commercial banks which could be beneficial through introduction of new managerial skills.

The mean of bank size was 7.567 ranging from a minimum of 5.143 to a maximum of 9.481. The standard deviation of bank size was 0.697. These results show that commercial banks' assets in Kenya varied within the period under study. The mean credit risk was 0.145 ranging from 0 to 4.523. This demonstrated a high degree of credit risk in commercial banks in Kenya with a standard deviation of 0.279. This implied that commercial banks in Kenya faced a challenge of non-performing loans which denied them net interest income.

## 4.2 Diagnostic tests

Prior to data analysis, it was vital to ensure that all the assumptions of regression models were met. Among the tests conducted were stationarity test, normality, heteroskedasticity, auto correlation and multicollinearity. Diagnostic tests results are shown in Table 3.

Results of Philips-Perron test in Table 3(a) indicated absence of unit root which implied that the data was stationary. The Doornik-Hansen test statistic for the multivariate normality in Table 3(b) indicated a significant $Chi^2$ statistic of 1.72e+05, which implied that data for the variables employed in the models was normally distributed. Table 3(c) shows that VIF values were less than 10 indicating absence of multicollinearity. In addition, Breuch-Pagan test in Table 3(d) show that data was homoskedastic (p-values > 0.05) and therefore appropriate for further analysis.

## 4.3 Ownership structures and net interest margin

The study sought to establish the relationship between ownership structures and net interest margin. Net interest margin was one of the financial performance measures used as a dependent variable. The results are presented in Table 4.

The results in Table 4 indicate a statistically significant $Ch^2$ statistic (545.08). This implied that the model was good for estimating NIM. The R-squared was 0.5304, implying that 53.04% of the variations in commercial banks' NIM were accounted for by the changes in ownership structure. Both institutional ownership and foreign ownership had statistically insignificant coefficients at 5% level of significance. This implied that institutional ownership and foreign ownership had an insignificant effect on NIM. In addition, Table 4 shows statistically significant coefficients of state ownership (-0.141) and managerial ownership (-0.093) respectively. These statistics indicated that state ownership and managerial ownership had a negative significant effect on commercial banks' NIM in Kenyan commercial banks. The results imply that a percentage increase in state ownership would lead to a decrease in banks' NIM by 14.1%. This finding confirms previous work by author in [38–40], who found that state owned banks exhibit lower financial performance associated with weaker corporate governance and high risk levels. In addition, the finding supported the works of [41, 42] who established a negative

**Table 3. Diagnostic tests.**

**(a) Stationarity statistics**

| Variable | z statistic | P-Value |
|---|---|---|
| NIM | -1.48 | 0.0231 |
| EPS | -5.17 | 0.000 |
| ROA | -1.73 | 0.0420 |
| ROE | -2.77 | 0.0027 |
| CR | 2.18 | 0.0342 |
| BS | -0.62 | 0.0262 |
| FO | 0.7391 | 0.0171 |
| IO | 1.8013 | 0.0451 |
| MO | 3.5782 | 0.0399 |
| SO | 1.1374 | 0.0423 |

**(b) Doornik-hansen test statistic**

| $Chi^2$ Statistic | P-Value |
|---|---|
| 1.72e+05 | 0.000 |

**(c) Multicollinearity**

| | VIF |
|---|---|
| Fitted values of NIM | **1.30** |
| Fitted values of EPS | **1.32** |
| Fitted values of ROA | **1.30** |
| Fitted values of ROE | **1.30** |

**(d) Breuch-pagan satistics**

| | Ownership Structure Variables | |
|---|---|---|
| | Test statistic | P Value |
| Variables: fitted values of NIM | 354.6 | 0.245 |
| Variables: fitted values of EPS | 532.9 | 0.167 |
| Variables: fitted values of ROA | 257.8 | 0.423 |
| Variables: fitted values of ROE | 438.7 | 0.156 |

relationship between state ownership and banks' financial performance. The findings of this study also support the work of [37] who found that ownership of commercial banks was ineffective due to weak monitoring efforts of government, political interference and weaker managerial incentives compared to privately owned banks especially in the Kenyan context. The

**Table 4. Ownership structure and net interest margin.**

| | Coef | Std. Err. | Z | P_value |
|---|---|---|---|---|
| SO | -0.141* | 0.051 | -2.74* | 0.006 |
| MO | -0.093* | 0.048 | -1.96* | 0.050 |
| IO | -0.048 | 0.034 | -1.39 | 0.165 |
| FO | 0.006 | 0.023 | 0.26 | 0.791 |
| BS | -0.034* | 0.013 | -2.53* | 0.011 |
| CR | 0.580* | 0.026 | 22.67* | 0.000 |
| cons | 0.285* | 0.105 | 2.7* | 0.007 |
| R-squared | 0.5304 | | | |
| Chi2 | 545.08* | | | 0.000 |

Dependent variable: NIM,

* indicates statistical significance at 5%

study finding contradicts the work of [43, 44] who found a positive relationship between state ownership and performance. Therefore, as suggested by [42], ownership structure reorganization through partial privatization need to be undertaken in those commercial banks with high level of government presence as a tool to improve performance.

Table 4 results indicate that a percentage increase in managerial ownership would lead to a decrease in banks' NIM by 9.3%. The finding of this study supports the work of [55] in United States who established a significant and negative relationship between managerial ownership and commercial banks financial performance. In addition, the results confirm the work of [56] in Australia who observed a negative relation between insider ownership and performance. This study therefore agrees with the earlier finding that management ownership decreases firms' performance associated with more voting powers by management controlling strategic decisions making tailored to their own personal gain [47].

The finding contradicts the work of [51] who found that management ownership is a structure that supports both the interest of shareholders and managers to enhance performance of a firm. This finding also contradicts the work of authors in [2, 32, 36, 48, 49, 50, 51] who found a positive relationship between managerial ownership and banks' performance.

## 4.4 Ownership structures and earnings per share

The study also sought to establish the relationship between ownership structures and earnings per share (EPS). The results are presented in Table 5.

Table 5 indicates a statistically significant $Ch^2$ statistic (20.72). This statistic implied that the model used to estimate EPS was good. The R-squared statistic was 0.0213. This indicated that 2.13% variations in commercial banks' EPS was as result of changes in banks' ownership structure. Table 5 results show statistically insignificant coefficients of state ownership (-92.26) and institutional ownership (-31.76). This implied that both state ownership and institutional ownership have insignificant effect on commercial banks' EPS. However, Table 5 shows a statistically significant coefficients of managerial ownership (-144.04) and foreign ownership (-72.62). These statistics implied that managerial ownership and foreign ownership had negative significant effect on banks' EPS. The results imply that a percentage increase in managerial ownership in Kenyan commercial banks would lead to a decrease in EPS by KES. 144.04. The study finding supports the earlier work of [47, 54] who observed that management ownership negatively relates to firms' performance. In addition, the study findings agree with the recent work by authors in [56, 57] who observed that insider ownership increases the controlling

**Table 5. Ownership structures and earnings per share.**

|  | Coef | Std. Err. | Z | P_value |
|---|---|---|---|---|
| SO | -92.26 | 75.71 | -1.22 | 0.223 |
| MO | -144.04* | 51.59 | -2.79* | 0.005 |
| IO | -31.76 | 44.79 | -0.71 | 0.478 |
| FO | -72.62* | 24.82 | -2.93* | 0.003 |
| BS | 25.31* | 10.68 | 2.37* | 0.018 |
| CR | 12.28 | 11.04 | 1.11 | 0.266 |
| cons | -86.41 | 80.18 | -1.08 | 0.281 |
| R-squared | 0.0213 |  |  |  |
| Chi2 | 20.71* |  |  | 0.002 |

Dependent variable: EPS,

* indicates statistical significance at 5%

powers of executives acting on their own interests. This executive act erodes shareholders' wealth in the long run. However, the finding contradicts the work of [2, 32, 36] who observed a positive relation between managerial ownership and banks' performance.

Results in Table 5 indicated that a percentage increase in foreign ownership in Kenyan commercial banks would lead to a decrease in EPS by KES 72.62. This study confirms the work of [75] who found that where foreign ownership does not play its monitoring roles effectively and prefer payment of dividends to improve corporate image of the firm, managers may represent information for their own interests. This leads to a decline in the firm's performance. The finding contradicts the work of [13] in China and [72] in Uganda and Botswana. These studies found that banks with foreign ownership performed better compared to those with absence of foreign ownership.

## 4.5 Ownership structures and return on assets

The study further sought to establish the relationship between ownership structure and return on assets. The model results are presented in Table 6. The $Ch^2$ statistic (333.54) was significant indicating that the model was fit for estimation of ROA. The R-squared statistic was 0.3137. This implied that 31.37% variations in commercial banks' ROA were accounted for by the changes in banks' ownership structure.

Table 6 indicates statistically insignificant coefficients of state ownership (-0.0521), managerial ownership (-0.0297) and foreign ownership (-0.0005). These statistics implied that state ownership, management ownership and foreign ownership had an insignificant effect on commercial banks' ROA. However, the institutional ownership coefficient (-0.0399) was statistically significant. This statistic indicated that institutional ownership had a negative significant effect on commercial banks' ROA. A one percentage increase in institutional ownership would lead to a decrease in ROA by 3.99%. The study finding supports the work of authors in [68, 69] who established a negative association between institutional ownership and financial performance. This finding contradicted the work of [51, 63, 65–67] who found a positive relationship between institutional ownership and firms' financial performance. In addition, the finding of this study contradicted the work of [3, 59] who observed that high firm's financial performance is deemed to be associated with institutional ownership which provides high quality management and improved corporate governance in the firms. This study associate's institutional ownership with low firm's performance. An increase in institutional ownership in commercial

**Table 6. Ownership structures and return on assets.**

|  | Coef | Std. Err. | z | P-value |
|---|---|---|---|---|
| SO | -0.0521 | 0.028 | -1.87 | 0.062 |
| MO | -0.0297 | 0.024 | -1.21 | 0.225 |
| IO | -0.0399* | 0.018 | -2.19* | 0.029 |
| FO | -0.0005 | 0.012 | -0.04 | 0.968 |
| BS | -0.0092 | 0.006 | -1.50 | 0.134 |
| CR | 0.1669* | 0.009 | 17.87* | 0.000 |
| cons | 0.0931* | 0.048 | 1.94* | 0.052 |
| R-squared | 0.3137 |  |  |  |
| Chi$^2$ | 333.54* |  |  | 0.000 |

Dependent variable: ROA,

* indicates statistical significance at 5%

**Table 7. Ownership structure and return on equity.**

|  | Coef. | Std. Err. | z | P_value |
|---|---|---|---|---|
| SO | -0.039 | 1.373 | -0.03 | 0.977 |
| MO | 3.861* | 1.305 | 2.96* | 0.003 |
| IO | 0.539 | 0.924 | 0.58 | 0.560 |
| FO | -0.523 | 0.634 | -0.82 | 0.409 |
| BS | 0.016 | 0.392 | 0.04 | 0.967 |
| CR | 1.276 | 0.915 | 1.39 | 0.163 |
| cons | -0.607 | 3.087 | -0.20 | 0.844 |
| R-squared | 0.0332 |  |  |  |
| Chi2 | 15.44* |  |  | 0.0171 |

Dependent variable: ROE,

* indicates statistical significance at 5%

banks especially where large institutional shareholders tend to make decisions for their own interests at the expense of minority stockholders results to a decline in financial performance.

## 4.6 Ownership structures and return on equity

Finally, the study sought to establish the relationship between ownership structures and return on equity. Results in Table 7 indicate a significant $Ch^2$ statistic (15.44) which implied that the model was good for estimation. Further, the R-squared statistic was 0.0332. This implied that 3.32% variations in ROE was as a result of changes in commercial banks ownership structure.

Table 7 indicates statistically insignificant coefficients of state ownership (-0.039), institutional ownership (0.539) and foreign ownership (-0.523). These statistics demonstrated that state ownership, institutional ownership and foreign ownership had an insignificant effect on commercial banks' ROE. The coefficient of management ownership (3.861) was statistically significant. A one percentage increase in management ownership would lead to an increase in commercial banks' ROE by 306.1%. This study finding supports the work of authors in [32, 36, 49] who observed a positive relation between management ownership and firms' performance. This study confirms the findings of [51] that management ownership is a tool for ownership structure that supports both the interest of shareholders and managers to enhance performance of a firm through an improved corporate governance practices. However, this finding contradicted the work of [55] in United States and other authors in [47, 52, 54, 56] who observed a negative relation between insider ownership and banks' performance.

## 4.7 Robustness check

To ensure reliability of the models' results, Hausman test was used to arrive at the appropriate models for data analysis. The study therefore made a choice between the fixed effects and random effect models. The null hypothesis under Hausman test was that the preferred model was random effect. Where the p-value was less than 0.05, the preferred model was random effects. The Hausman test results presented in Table 8 indicate a significant $Chi^2$ statistics for the dependent variables. This implied that the preferred model was random effect for each of the four dependent variables employed in the study. Robustness of results was also checked using maximum likelihood (ML) model. The ML results in Table 9 showed that the results were robust. Most of the variables retained their coefficient signs and statistical significance.

**Table 8. Hausman test results.**

| Dependent variable | Chi$^2$ | p-value | Preferred model |
|---|---|---|---|
| NIM | 55.93 | 0.000 | Random effects |
| EPS | 14.94 | 0.021 | Random effects |
| ROA | 81.12 | 0.000 | Random effects |
| ROE | 25.04 | 0.003 | Random effects |

**Table 9. Random-effects ML regression results.**

| | NIM | EPS | ROA | ROE |
|---|---|---|---|---|
| SO | -0.145(-2.53)* | -92.28(-1.23) | -0.057(-1.53) | -0.039(-0.03) |
| MO | -0.096(-1.83) | -144.36(-2.64)* | -0.024(-0.80) | 3.862(2.98)* |
| IO | -0.056(-1.40) | -31.81(-0.71) | -0.061(-2.22)* | 0.539(0.59) |
| FO | 0.005(0.19) | -72.71(-2.88)* | 0.001(0.04) | -0.523(-0.83) |
| BS | -0.042(-2.35)* | 25.33(2.37)* | -0.018(-2.21)* | 0.016(0.04) |
| CR | 0.584(22.67)* | 12.28(1.12) | 0.171(18.37)* | 1.276(1.41) |
| Chi$^2$ | 358.67* | 19.94* | 248.4* | 15.29* |

## 5. Summary and conclusions

This study examined the relationship between ownership structures (state ownership, managerial ownership, institutional ownership and foreign ownership) and financial performance (net interest margin, earnings per share, return on assets and return on equity) for a sample of 39 commercial banks in Kenya, for the period 2009 to 2020. The findings indicated a strong evidence of ownership structures explaining the differences in commercial banks' financial performance in Kenya. The results established that the greatest influence of ownership structures was on NIM (53.04%), followed by ROA (31.37%). The influence of ownership structures was low on ROE (3.32%) and EPS (2.13%).

The results found a significant and negative relationship between state ownership and NIM. However, state ownership was found to have an insignificant effect on both EPS, ROA and ROE. The expectation was that state ownership would improve governance practices for improved performance. The study concludes that state ownership of commercial banks is ineffective due to weak monitoring efforts of governments resulting, weaker managerial incentives and political interference that interfere with corporate governance practices. As a result, ownership structure reorganization through partial privatization is essential in those commercial banks with high level of government presence as a tool for improving bank financial performance.

The findings showed a significant and negative relationship between management ownership and both NIM and EPS. The findings also established a significant and positive relationship between management ownership and ROE. The expectation of this study was that management ownership would positively relate with banks' financial performance. However, in the contrary, evidence established a negative relationship between NIM and EPS.

The study results established insignificant effect of institutional ownership on NIM, EPS and ROA. However, the results found evidence that institutional ownership has a significant and negative effect on ROA. This study associates poor financial performance in commercial banks especially in Kenya with high level of institutional ownership. Large institutional shareholders tend to make decisions for their own interests at the expense of minority and individual stockholders resulting to a decline in financial performance.

Lastly, this study established an insignificant relationship between foreign ownership and NIM, ROA and ROE, but a significant and negative relationship between foreign ownership and EPS. The expectation was that the presence of foreign ownership would result to an increase in financial performance. The finding of this study was contrary as foreign ownership had no effect on NIM, ROA and ROE. In addition, presence of foreign ownership resulted to a decline in EPS. This was attributed to low levels of monitoring by foreign investors.

## 5.1 Recommendations

Based on the above conclusions, the findings of this study have the following practical implications. First, the results show that ownership structures influence financial performance of commercial banks in Kenya. The most affected financial performance was NIM and ROA. Thus, the finding of this study suggests that regulators of commercial banks should require all commercial banks to vary their ownership structures optimally to boost financial performance of commercial banks. Secondly, study found a significant negative association between state ownership and NIM. On the other side, an insignificant effect of state ownership was observed on EPS, ROA and ROE. The study concluded that there was ineffectiveness of state ownership due to weak monitoring efforts of governments. This study recommends that banks with high percentage of state ownership should consider full or partial privatization to improve corporate governance practices which is essential for improving banks' performance. Third, the study found a significant negative association between management ownership and both NIM and EPS and a positive association between management ownership and ROE. The study concluded that the existence of negative association was possibly because bank managers control high level of equity gaining more powers to influence strategic decisions that benefit them instead of aiming at improving performance positively. Policy makers should adopt managerial ownership policy limiting the proportion of equity bank managers should hold. This would limit their powers and put them under monitoring by other equity holders.

Fourth, the study also found that institutional ownership had a significant negative relationship with ROA. The negative relationship could be attributed to the large proportion of institutional shareholding. As a result, high institutional shareholding especially by few institutional investors tends to make decisions for their own gain at the expense of improving corporate governance. Since institutional ownership had no influence on other financial performance measures, bank executives should ensure optimal proportion of institutional ownership to avert its negative effect on ROA. The study also suggests that a percentage limit of equity stock an institutional investor should hold be put in place. Lastly, the study expected that foreign ownership would influence banks' performance positively. However, evidence from this study found an insignificant effect between foreign ownership and NIM, ROA and ROE and a significant negative relationship between foreign ownership and EPS. The study concluded that this finding was as a result of low level of monitoring by foreign investors on banks' operations. The study therefore suggests that bank regulators should come up with a policy detailing the role and place of foreign investors in strategic decision making. This will ensure that their presence is felt in every decision undertaken by bank managers.

## 5.2 Suggestion for further study

This study had some limitations. First, the study considered only one aspect of corporate governance (ownership structure) in commercial banks. The study therefore suggests a further study where other corporate governance aspects such as board size, board independence, board skills are all included. Secondly, the study focused on the Kenyan commercial banking sector. A similar study could be carried out covering a wider region especially the East African

Community and the Inter-governmental Authority on Development Region. Despite these limitations, the findings of this study contributes to both practice and existing literature on ownership structure.

## Supporting information

**S1 File.**
(PDF)

## Author Contributions

**Conceptualization:** Peter Njagi Kirimi.

**Data curation:** Peter Njagi Kirimi.

**Formal analysis:** Peter Njagi Kirimi.

**Methodology:** Peter Njagi Kirimi, Samuel Nduati Kariuki, Kennedy Nyabuto Ocharo.

**Supervision:** Samuel Nduati Kariuki.

**Validation:** Samuel Nduati Kariuki, Kennedy Nyabuto Ocharo.

**Writing – original draft:** Peter Njagi Kirimi.

**Writing – review & editing:** Samuel Nduati Kariuki, Kennedy Nyabuto Ocharo.

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
