## [Decision Letter · Decision Letter 0]

23 Mar 2022

PONE-D-21-35480Ownership structure and financial performance: Evidence from Kenyan commercial banks.PLOS ONE

Dear Dr. Peter Njagi Kirimi,

Thank you for submitting your manuscript to PLOS ONE. After careful consideration, we feel that it has merit but does not fully meet PLOS ONE’s publication criteria as it currently stands. Therefore, we invite you to submit a revised version of the manuscript that addresses the points raised during the review process.

We look forward to receiving your revised manuscript.

Kind regards,

Ricky Chia Chee Jiun

Academic Editor

PLOS ONE

Journal Requirements:

Additional Editor Comments (if provided):

Reviewers' comments:

Reviewer's Responses to Questions

5. Review Comments to the Author

Reviewer #1: It was understood that a lot of effort was spent on this study. The topic is current. Literature research is sufficient. English level is appropriate. Although the number of pages is high, it is appropriate when the whole study is taken into account. However, some mistakes in the citations should be corrected. For example, the reference to Kenya's central bank on page 7 is dated 2016. It is not in this date in bibliography. On page 8, "Amneh Amneh, Hussam & Mahmoud, 2021" also missed in the references. . In addition, in some citations the authors are written individually, in some of them "et al." used. One format of citations should be preferred.

---

## [Author Response · Author response to Decision Letter 0]

7 Apr 2022

Comments to the editor comments:

The whole paper was aligned to meet PLOS ONE style requirement. The references were reviewed and aligned to PLOS One Style requirements. Correction matrix file attached.

Comments to the Reviewer:

All in-text citations and references were rectified as per the PLOS one requirements.

---

## [Editor Report · Decision Letter 1]

27 Apr 2022

Ownership structure and financial performance: Evidence from Kenyan commercial banks.

PONE-D-21-35480R1

Dear Dr. Peter Njagi Kirimi,

We’re pleased to inform you that your manuscript has been judged scientifically suitable for publication and will be formally accepted for publication once it meets all outstanding technical requirements.

Kind regards,

Ricky Chee Jiun Chia

Academic Editor

PLOS ONE
---

## [Editor Report · Acceptance letter]

29 Apr 2022

PONE-D-21-35480R1 

Ownership structure and financial performance: Evidence from Kenyan commercial banks. 

Dear Dr. Kirimi:

I'm pleased to inform you that your manuscript has been deemed suitable for publication in PLOS ONE. Congratulations! Your manuscript is now with our production department. 

Kind regards, 

on behalf of

Dr. Ricky Chee Jiun Chia 

Academic Editor

PLOS ONE